# First Detection of *Salmonella enterica* Serovar Choleraesuis in Free Ranging European Wild Boar in Sweden

**DOI:** 10.3390/pathogens11070723

**Published:** 2022-06-24

**Authors:** Linda Ernholm, Susanna Sternberg-Lewerin, Erik Ågren, Karl Ståhl, Cecilia Hultén

**Affiliations:** 1Department of Biomedical Sciences and Veterinary Public Health, Faculty of Veterinary Medicine and Animal Science, Swedish University of Agricultural Sciences (SLU), SE-750 07 Uppsala, Sweden; linda.ernholm@sva.se; 2Department of Disease Control and Epidemiology, National Veterinary Institute (SVA), SE-751 89 Uppsala, Sweden; karl.stahl@sva.se (K.S.); cecilia.hulten@sva.se (C.H.); 3Department of Pathology and Wildlife Diseases, National Veterinary Institute (SVA), SE-751 89 Uppsala, Sweden; erik.agren@sva.se

**Keywords:** wildlife/livestock interface, surveillance, *Salmonella* Choleraesuis, wild boar, *Sus scrofa*

## Abstract

Following the first detection of *Salmonella enterica* subsp. *enterica*, serovar Choleraesuis (*S*. Choleraesuis) in a Swedish pig herd for more than 40 years and subsequent detection of the same serotype in an enclosure with kept wild boar, a national surveillance for *S.* Choleraesuis in free living wild boar was launched. A total of 633 wild boar sampled within the active and the enhanced passive surveillance were examined for *Salmonella enterica* serovars by culture. Of these, 80 animals were culture positive for *S*. Choleraesuis var. Kunzendorf. All positive animals, including those in the original outbreaks, originated from counties located in the southern and eastern parts of Sweden. Fifty-eight isolates were selected for sequence typing, revealing a relatively homogenous population of *S*. Choleraesuis with two distinct genetic clusters containing isolates from the southern counties in one and the counties further northeast in the other. Sequenced isolates from domestic pig farms all clustered with wild boar in the same region. *S.* Choleraesuis appears highly contagious in dense wild boar populations, making it a relevant model for other infectious diseases that may be transmitted to pigs. The many potential routes of introduction and spread of *S*. Choleraesuis warrant further investigations in order to prepare for other disease threats.

## 1. Introduction

Many contagious diseases such as African swine fever (ASF), classical swine fever (CSF), Aujeszky’s disease (AD), and porcine reproductive and respiratory syndrome (PPRS) are absent in the Swedish pig population [1]. The last outbreak of CSF was in 1944, and AD was eradicated in 1996. PRRS was detected for the first time in 2007 but eradicated shortly thereafter [2]. ASF has never been detected in the country, but the spread within Europe and the role of European wild boar (*Sus scrofa*) is a continuous worry for Swedish pig producers. Despite biosecurity programs in pig holdings (including all-in-all-out indoor production, with hygiene locks at building entrances), the risk of disease transmission between wild boar and domestic pigs has increased due to growth of the wild boar population, and the transmission of other viruses between domestic pigs and wild boar in Sweden has been demonstrated [3].

The importance of longitudinal surveillance of diseases in wildlife has been highlighted in many studies as reviewed by Barroso et al. (2022) [4]. The Swedish general wildlife disease surveillance program, based on passive surveillance of animals found dead, has been in place since the 1940s [5]. This program has contributed to baseline knowledge of diseases present in the wildlife population and provided a large sample collection, which has allowed for retrospective investigations of certain diseases. Within the surveillance program, all wildlife species are tested for *Salmonella* upon suspicion. Moreover, an enhanced passive surveillance of ASF in wild boar has been implemented since 2013 [1].

In the 18th century, the free-living wild boar population was eradicated in Sweden. In the 1970s, a few wild boar escaped their fences in hunting estates in the Southern part of the country and became part of the wild fauna. In 1981, a decision was taken to reduce the population to below 100 animals, but this was later revoked, and since the late 1980s, the population has grown steadily [6]. The national hunting bag has been around 120,000 animals/year during the past five years [7], and the total population was estimated to be at least 300,000 in 2020 [6]. Wild boar are present in all counties in the southern parts of Sweden, where, in some areas, a high population density coincides with the location of pig holdings (Figure 1a,b), emphasizing the need for disease surveillance in the wild boar population.

A national *Salmonella* control program was initiated in the 1950s and 1960s and was gradually developed to its current form, covering the entire chain from feed to food. This program was the basis for the additional guarantees regarding *Salmonella* when Sweden joined the European Union in 1995. These guarantees allow national requirements for *Salmonella* sampling of fresh meat from cattle, pigs and poultry, table eggs, and raw feed materials brought into Sweden. The program focuses on food-producing animals with the objective of *Salmonella*-free products originating from domestic livestock.

In 2020–2021, *Salmonella enterica subsp. enterica*, serovar Choleraesuis (*S*. Choleraesuis) was detected in five domestic pig herds and one estate with a small group of fenced wild boar. These were the first findings of this serovar in more than 40 years [1]. Similar to human infections with *S*. Typhi and Paratyphi, *S*. Choleraesuis is a pig-adapted serovar that can cause a clinical picture resembling swine fever, and a high mortality may be seen in domestic pig herds [8], particularly in the presence of other infections [9]. Historically, it was the most common serotype in pigs worldwide but is now rarely seen in domestic pigs in Europe [10]. A study on 102 isolates from Europe and the United States used molecular epidemiology to reveal geographical clustering of isolates and a possible association with poorly disinfected vehicles in outbreaks in Danish pig holdings [10]. Detailed study of isolates from the Danish outbreaks also indicated several introductions and a possible link to corn transported from Eastern Europe [9]. The bacteria can survive for long periods in the environment and have been shown to persist in dry feces from infected pigs for up to 13 months [11].

Human infections are rare but may be severe, due to the systemic nature of the infection presenting as septicemia, mostly in young or debilitated individuals [8,12].

In wild boar, the clinical signs of infection with *S*. Choleraesuis appear similar to those in domestic pigs [13]. Molecular typing of isolates from an outbreak in Italian wild boar could not detect a link to isolates from domestic pigs [14], while a German study revealed different genetical clusters of wild boar isolates, of which one also included isolates from domestic pigs [15]. Indications of an increased prevalence of *S*. Choleraesuis among wild boar have been noted in Germany, possibly associated with a heightened awareness of the ASF risk, leading to more post-mortem examinations of wild boar [16]. Although transmission patterns differ slightly, the similarities between infection with *S*. Choleraesuis and ASF infer that close study of *S*. Choleraesuis outbreaks in wild boar may provide useful knowledge for the surveillance and control of ASF.

After the detection of *S.* Choleraesuis in domestic pigs in Sweden, surveillance targeting this agent in free-ranging wild boar was initiated, to complement the wildlife disease surveillance program. The design and results from the surveillance of wild boar since the first detection of *S.* Choleraesuis are described in this report.

## 2. Results

A total of 633 wild boar sampled within the active and the enhanced passive surveillance were examined for *Salmonella enterica* serovars by culture. Of these, 80 animals were culture positive for *S*. Choleraesuis var. Kunzendorf (Figure 1c, Table 1) in at least one of the materials collected from each animal (Table 2). All positive animals, including those in the original outbreaks, originated from counties located in the southern (Skåne and Halland) and eastern (Södermanland, Stockholm, and Östergötland) parts of the country (Figure 1c and Figure 2).

A total of seven *Salmonella* serotypes other than *S*. Choleraesuis were detected including *S*.Diarizonae (nine); *S*. Typhimurium (four); *S*. Newport (two); and one of each of *S.* Hessarek, *S*. Duesseldorf, *S*. Enteriditis, and *S*. Coeln. In addition, one isolate was identified of antigen type ‘O4′ and four of antigen type ‘O6,8′, with no further serotyping available.

The detection of *S.* Choleraesuis was significantly (*p* < 0.01) more frequent from the carcasses of wild boar found dead than from wild boar sampled at hunting. This association between category and detection was not seen for other *Salmonella* serotypes in this study.

### 2.1. Wild Boar Found Dead

In this category, a total of 100 wild boar were sampled with one to four materials each, depending on availability and suitability. For 14 of these animals, the collected sample materials (*n* = 2−3) were analyzed as individual pools (i.e., one from each animal), all with negative results. The results from each type of individually cultured sample material are shown in Table 2. Of the 100 animals, 27 were culture positive for *S.* Choleraesuis, and, with two exceptions, all sample materials from these animals were positive. One of the 27 was, in addition to *S*. Choleraesuis, also positive for another *Salmonella*, while three animals of the 100 were positive for *Salmonella* of other serotypes only.

The sex of the wild boar was recorded for 77 of the animals (Table 3). Although *S*. Choleraesuis was isolated from more female than male animals, the association was not significant (*p* = 0.10). There was no obvious association between the detection of *S.* Choleraesuis and the age category of the animal among the wild boar found dead.

### 2.2. Samples from Hunted Wild Boar

A total of 533 wild boar were sampled by hunters, at normal hunting. For 448 of these, information about the sex of the animal was provided, and 46% were male and 54% female. While both requested materials, a mesenteric lymph node (MLN) and a fecal sample, were submitted from 509 animals, only the fecal sample was available from 20 animals, and from four animals, just the MLN was available. Both materials were available from 51 out of 53 wild boar from this category that were positive for *S*. Choleraesuis. Out of these, 12 (23.5%) were positive in both MLN and feces, 17 (33.3%) only in the mesenteric lymph node, and 22 (43.1%) in feces alone. All *S*. Choleraesuis positive wild boar among the hunter collected samples were shot in the before-mentioned counties of Skåne, Södermanland, and Stockholm, and the proportion of positives did not differ between the sexes. However, the proportion of young animals with positive culture results was significantly higher than for adult animals (*p* < 0.01).

### 2.3. Sequencing

When the surveillance was initiated, isolates previously detected in the wildlife dis-ease surveillance but not fully typed were re-examined and sequenced. Two isolates from the most southern area, one from 2018 and one from June 2020, were identified as *S*. Chol-eraesuis and included in the sequence typing, together with a selected number of isolates from the current surveillance.

All selected isolates were confirmed by whole-genome sequencing to be multi-locus sequence type (ST) 145, consistent with *S*. Choleraesuis var. Kunzendorf [17].

Whole-genome sequencing revealed a relatively homogenous population of *S*. Choleraesuis; among 58 sequenced isolates from 2020–2022, there were only a total of 96 SNPs, most of which were unique for individual isolates or small groups (Figure 3). Isolates clustered by hunting district, however, not consistently so. A genetic separation between isolates from the southern (Skåne and Halland) counties and the counties further northeast was evident, although based on very few SNPs. Sequenced isolates from three pig farms in Skåne county all clustered with wild boar in the same region. A comparison of the Swedish 2020–2022 sequence cluster to publicly available sequences in EnteroBase revealed a high degree of similarity to wild boar isolates from central Europe, including Poland, Germany, and the Czech Republic (HierCC HC50 79087).

The isolate from 2018 did not show genetic similarity with the isolates in the current outbreak, while the isolate from June 2020 is seen centrally in the “Skåne county” cluster in Figure 3.

## 3. Discussion

The long-standing wildlife disease surveillance and *Salmonella* control programs in Sweden have provided a historical context supporting the assumption of a recent introduction of *S*. Choleraesuis. An established collaboration with the hunters’ organizations allowed for rapid enrolment in the voluntary sampling effort.

When relying on samples from hunter-harvested animals from the ordinary hunting bag, similar to most active surveillance, the collection will have an element of convenience sampling and inherently consist of apparently healthy animals. By complementing the samples from wild boar found dead with sourcing samples from apparently healthy, hunted wild boar in varying locations, our sampling strategy provided as far a representative picture as possible of *S*. Choleraesuis in the wild boar population. As the aim in this study was disease detection rather than national prevalence estimation, the samples from wild boar found dead were useful as a risk-based sampling. In order to strengthen the assessment of the future probability of introduction to domestic pig herds, more detailed prevalence figures in combination with data on potential transmission routes between wild boar and domestic pigs would be needed. Nevertheless, our data indicate that transmission between wild boar and domestic pigs may be a significant factor in the spread of *S*. Cholerasuis in the Swedish context and that this route of introduction to domestic pigs might be relevant for a number of infectious diseases. In this context, studies on *S*. Choleraesuis in affected areas may serve as a proxy for ASF and contribute to preparedness for ASF outbreaks in new regions. In addition, surveillance samples that are negative for ASF should be examined for *S*. Choleraesuis, in regions where this infectious agent has not previously been detected.

In the latest European Union One Health 2020 Zoonoses Report (p. 75) [18], only four countries reported *Salmonella* in wild boar. Some studies have reported the finding of *Salmonella* antibodies or PCR reactions in wild boar in Scandinavia [19,20], while *S*. Choleraesuis in wild boar has previously been reported from Italy [14,21], Germany [16], Austria, France, Estonia and Hungary [10], and Spain [13,22]. The risk of human infection via pork products is acknowledged but, based on reported numbers of human cases, appears to be less common than other serotypes [18]. We have not been able to obtain information about the presence of human cases in the 1970s, when *S*. Choleraesuis was present in domestic pigs in Sweden. As the wild boar population at that time was almost extinct, the current situation is new. Since the prevalence of *S*. Choleraesuis in wild boar may be high in densely populated areas, as reflected in our study, it could present a public health risk via consumption of meat products from infected wild boar. During the outbreak, the Swedish Food Administration presented a scientific opinion on *S*. Choleraesuis from wild boar and disseminated advice on relevant food hygiene aspects to hunters and the public.

To assess the probability of foodborne disease, sampling of apparently healthy animals is needed, as these reflect the population of interest. However, these animals would not be expected to have an established systemic infection, and hence, selecting the optimal sample material is a challenge. In this study, samples consisting of lymph node and feces were collected. Roughly one-third of the *Salmonella*-positive wild boar were positive in both materials, one-third in just the lymph node, and one-third only in feces. Hence, when testing apparently healthy wild boar, it is advisable to sample at least two materials to increase the probability of detection.

The sometimes-extended period between death of the animal and sample analysis may lead to bacterial overgrowth and impair the detection of *S*. Choleraesuis. This aspect would be most relevant for the sampling of wild boar found dead, but as these animals are expected to have died from septicemia, the bacteria will be present in high numbers in many organs, and hence, detection may still be possible. The fact that *S*. Choleraesuis causes systemic disease and death among wild boar makes sampling of wild boar found dead a logical approach for disease detection in new areas.

The origins of outbreaks of wildlife disease are difficult to investigate. In the light of previous reports from Denmark [9], indicating a possible introduction via corn from Eastern Europe, this route of introduction to Sweden is not entirely unlikely. We know from other studies (unpublished data) that feeding of wild boar with imported corn from Eastern Europe is not uncommon in Sweden; however, no such feed has been available for sampling. The affected areas are characterized by dense populations of wild boar and the presence of hunting estates with both regular feeding activities and regular visits from international hunters. Despite genetic clustering according to geographic origin within Sweden, the isolates are not so different as to indicate numerous different introductions, at least not from different regions. The sequencing results demonstrate similarities with strains from Poland, Germany, and Czech Republic, indicating a possible connection with these countries. In addition, the low variation between the Swedish isolates indicates a rather recent introduction. The outbreaks in domestic pig herds were most likely caused by spillover from the wild boar population.

As many important pig diseases that are currently absent in Sweden can be established and spread in wild boar populations, this outbreak may serve as a warning and an opportunity to investigate how a very low probability of introduction for individual events may still, eventually, result in an established disease outbreak. The many potential routes of introduction and spread of *S*. Choleraesuis warrant further investigations in order to address other disease threats.

## 4. Materials and Methods

The surveillance activity was designed by applying a combination of enhanced passive and active surveillance. Data on GIS-coordinates and the sex and estimated age of the wild boar were collected via the submission form. When possible, all wild boar were sampled by a mesenteric lymph node (MLN) and a fecal sample.

### 4.1. Wild Boar Found Dead

Wild boar found dead and submitted for necropsy within the wildlife disease surveillance program, as well as material from wild boar found dead and sampled in the field within the ongoing surveillance for African swine fever, were cultured for salmonellae. Due to cadaverous changes or missing organs, materials other than the above-mentioned were sometimes used, including muscle, blood-bearing organs or bone marrow.

### 4.2. Samples from Hunted Wild Boar

Appearingly healthy wild boar were sampled during hunting in the period beginning October 2020 to the end of February 2022. Sampling kits were assembled and dispatched from the National Veterinary Institute to hunter organizations and hunters that volunteered to assist in sampling in areas of geographic interest. Initially, these were areas around the detected cases but later expanded to all counties with a known wild boar population. From hunter-harvested wild boar, a mesenteric lymph node and a fecal sample was collected.

### 4.3. Bacterial Culture

Sample materials submitted were individually cultured for *Salmonella enterica* serovars in accordance with EN-ISO 6579-1:2017. Briefly, this included pre-enrichment in buffered peptone water followed by culture on MSRV (Modified Semi-solid Rappaport Vassiliadis) agar plates at 41.5 °C for 24–48 h and subsequent culture of suspect colonies on XLD (xylose-lysine-deoxycholate) and BG (Brilliant Green) agar at 37 °C for 24 h. All suspect isolates were tested for O- and H-antigen, and positive isolates were further classified using the White–Kaufmann–Le Minor scheme. Strains with O6,7:c:1,5 or O6,7:-:1,5 were further biochemically tested using H2S and Dulcitol., with all isolates being H2S+ and Dulcitol–, which is compatible with var. Kunzendorf.

### 4.4. Sequencing

DNA was extracted from cultures of selected isolates using the IndiMag Pathogen kit (Indical Bioscience GmbH, Leipzig, Germany) on a TANBead Maelstrom-9600 automated system and quantified using the Qubit BR dsDNA kit (Thermo Fisher, Waltham, MA, USA). Library preparation was carried out using Nextera chemistry, and sequencing was performed using either an Illumina NovaSeq instrument at SciLifeLab Clinical Genomics, Solna, Sweden, or an in-house Illumina MiSeq instrument. All isolates were sequenced to a minimum of 40× coverage. Sequence data and relevant metadata are available at the European Nucleotide Archive [23] under project accession PRJEB52916. Genetic distances between isolates were determined by single-nucleotide polymorphism (SNP) analysis as previously described [24] and visualized with the NeighborNet algorithm in the open software SplitsTree 4.14.4. A comparison with publicly available genome sequences of S. Choleraesuis isolates from other countries was done by core-genome multi-locus sequence typing (cgMLST) including HierCC hierarchical clustering in EnteroBase (https://enterobase.warwick.ac.uk/, accessed on 28 May 2022).

Geographical maps were produced in the statistical open-source software ‘R’ (R Core Team, 2021, Vienna, Austria) based on data on shot wild boar from the Swedish Hunters’ association and the Swedish board of Agriculture regarding the pig enterprises.

Statistical analyses were done in the statistical open-source software ‘R’ (R Core Team, 2021, Vienna, Austria), with the addition of the package ‘tidyverse’ [25]. Associations between two variables were assessed by Pearson’s chi-squared test or Fisher’s exact test.

Results from individual wild boar were communicated to the submitter, and aggregated results were visualized in an interactive map on the website of the National Veterinary Institute. Any personal data were handled according to GDPR within the laboratory information system of the National Veterinary Institute.

## 5. Conclusions

*S*. Choleraesuis appears highly contagious in dense wild boar populations, making it a relevant model for other infectious diseases that may be transmitted to pigs.

Wild boar found dead constitute a useful source of sampling material, but sampling hunted animals can also be applied in surveillance. In the latter case, both the mesenteric lymph node and feces are recommended to increase the probability of detection.

## Figures and Tables

**Figure 1 pathogens-11-00723-f001:**
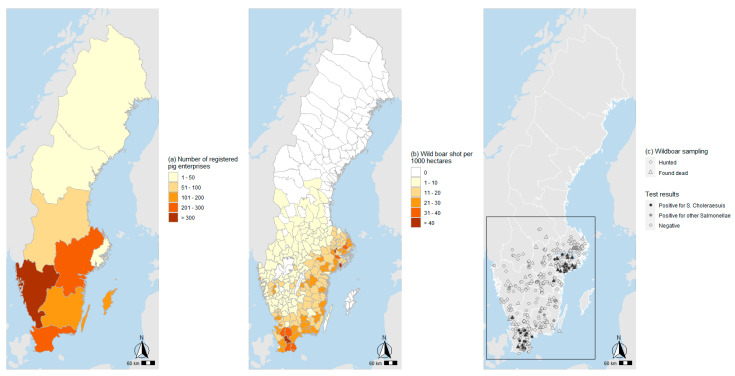
(**a**–**c**) Geographic distribution of Swedish pig holdings (**a**); wild boar population, based on hunting bags (**b**) and sampled wild boar in this study (**c**). The square indicates the area shown in higher resolution in Figure 2.

**Figure 2 pathogens-11-00723-f002:**
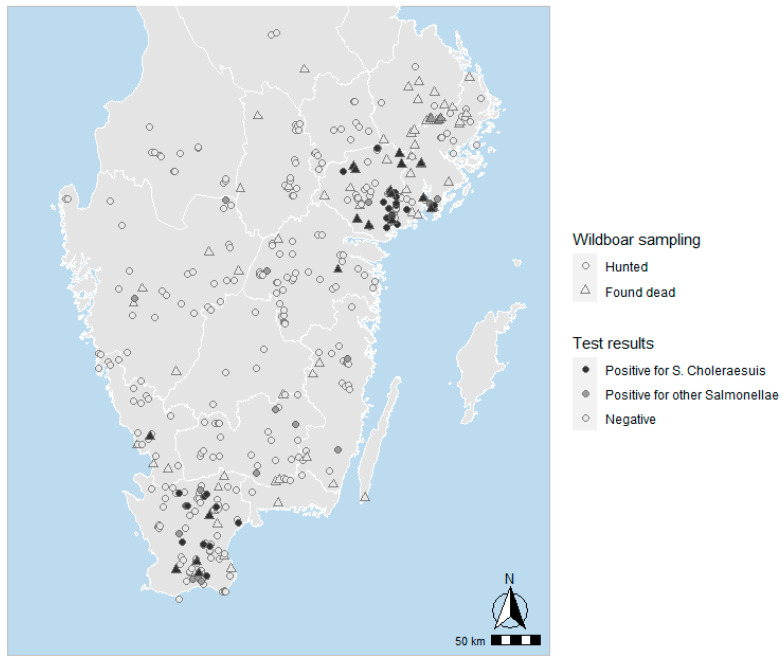
Geographic location of sampled wild boar and results of sample analyses in the surveillance of Swedish wild boar for *Salmonella* enterica serovars in 2020–2022.

**Figure 3 pathogens-11-00723-f003:**
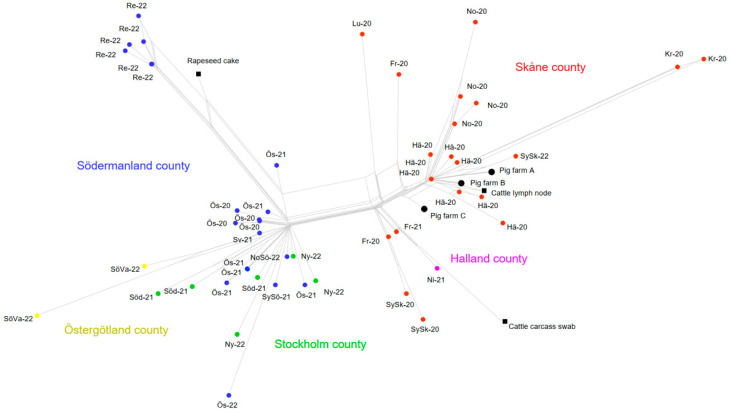
NeighBor net visualization of whole-genome SNP variation among all sequenced isolates of *S*. Choleraesuis in 2020–2022. Isolates are labelled by hunting district and year and colored according to the county of the hunting district.

**Table 1 pathogens-11-00723-t001:** Results from testing wild boar for *S.* Choleraesuis.

Surveillance Category	Positive for *S*. Choleraesuis	Negative for *S*. Choleraesuis
Active, hunted	53	480
Passive, found dead	27	73

**Table 2 pathogens-11-00723-t002:** Results from sample materials cultured individually from Swedish wild boar found dead during 2020–2022.

Material (*n*)	*S*. Choleraesuis	Other Salmonellae
Mesenteric lymph node (52)	34.6%	3.8%
Intestine (37)	43.2%	2.7%
Feces (24)	20.8%	4.2%
Bone marrow (22)	18.2%	0
Tonsil (10)	10.0%	20.0%
Spleen (11)	54.5%	0
Liver (1)	0	0
Muscle (2)	50.0%	0
Stomach (1)	100%	0
Kidney (1)	0	0
Joint (1)	100% *	100% *

* The joint sample from one animal yielded both *S*. Choleraesuis and *S*. Newport.

**Table 3 pathogens-11-00723-t003:** Recorded sex of wild boar found dead and their status for *S.* Choleraesuis.

Sex	Neg. for *S.* Choleraesuis	Pos. for *S.* Choleraesuis
Male	31	5
Female	27	14

## Data Availability

Upon reasonable request, anonymized data from the surveillance are available from the corresponding author. Raw data for the sequenced strains were uploaded to European Nucleotide Archive (https://www.ebi.ac.uk/ena, accessed on 28 May 2022) under project accession PRJEB52916.

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
