# Peer review of "First Detection of Salmonella enterica Serovar Choleraesuis in Free Ranging European Wild Boar in Sweden"

_pathogens, 2022, doi:10.3390/pathogens11070723_

Round 1

Reviewer 1 Report

Ernholm et al. submitted their manuscript entitled ‚First detection of Salmonella enterica serovar Choleraesuis in free ranging European wild boar in Sweden '. The authors paid attention to the appearance of S. Choleraesuis in wild boars in southern and western Sweden. They describe their results and methods, including different attitudes to samples from spontaneously dead or hunted boars.

L12: It should be introduced that Salmonella enterica serovar Choleraesuis will be mentioned as S. Choleraesuis.

L66-68: The popularity of pigs as animal models for human diseases has increased (doi: 10.1126/scitranslmed.abd5758). I think it would be suitable to compare it to human diseases too. While S. Typhimurium causes in both previously healthy humans and pigs salmonellosis (enterocolitis), S. Cholerasuis causes in pigs typhoid-like fever caused in humans with S. Typhi and Paratyphi.

L94-96: A total number of Salmonella serovars would be interesting information.

L94: Salmonella spp.? There are two species S. enterica and S. bongori (doi: 10.3389/fimmu.2014.00481), but S. enterica serovars only are important enteric pathogens of humans, livestock, poultry, and warm-blooded pets. S. bongori is a reptile pathogen. It probably is better to write "S. enterica serovars ".

L235-298: Material and Methods (MM) should be described factually and shortly. The text of MM in some of its parts looks more like a Discussion than MM. Some descriptions of methods have already been described in Results and Discussion. Please, modify texts and prevent their duplication. Moreover, MM should be subdivided according to methods/approaches. It will improve the arrangement of the used methods similarly, as it was done in the description of the Results.

Author Response

Thank you for your comments See uploaded file for response

Reviewer 2 Report

Dear Authors,

your study is interesting and very well planned. The interaction between hunters and National Veterinary Service you described in the paper is of the greatest importance and should be applied in other countries to investigate on wildlife diseases.

My comments are the following:

1.      Which biosecurity measures are commonly implemented in Swedish pig herds?

2.      How do you explain the “escaping” of S. Choleraesuis from pig herds into wild boar population, being biosecurity measures in place? This escaping is not so common, even in areas with high density of pigs and wild boars (see Bonardi et al., EcoHealth volume 16, pages 420–428 (2019)).

3.      In the Results section, you mentioned some “tested material” without describing it (L 95).  Please, add a proper description.

4.      At line 112, you mentioned “carcasses”, and I’m sure you were referring to carcasses of animals found dead. Please, specify properly.

Minor corrections:

Salmonella must be written in Italics and with capital letter (Salmonella) throughout the manuscript. See lines 45, 57, 59, 60, 63, 165188,189, 205

Salmonellae must not be written in Italics: see lines 105 and 240. No capital letter for “salmonellae”.

S. must be written in Italics: see lines 106, 128, 130, and 213

Author Response

(The authors gave the same response as above.)
